



# A renewed rise in global HCFC-141b emissions between 2017-2021

Luke M. Western[1,2], Alison L. Redington[3], Alistair J. Manning[3], Cathy M. Trudinger[4], Lei Hu[1,5],
Stephan Henne[6], Xuekun Fang[7], Lambert J.M. Kuijpers[8], Christina Theodoridi[9], David S. Godwin[10],
Jgor Arduini[11], Bronwyn Dunse[4], Andreas Engel[12], Paul J. Fraser[4], Christina M. Harth[13], Paul
B. Krummel[4], Michela Maione[11], Jens Mühle[13], Simon O'Doherty[2], Hyeri Park[14], Sunyoung Park[14],
Stefan Reimann[6], Peter K. Salameh[13], Daniel Say[2], Roland Schmidt[13], Tanja Schuck[12], Carolina Siso[1,5],
Kieran M. Stanley[12], Isaac Vimont[1,5], Martin K. Vollmer[6], Dickon Young[2], Ronald G. Prinn[15], Ray
F. Weiss[13], Stephen A. Montzka[1], and Matthew Rigby[2]

[1]Global Monitoring Laboratory, National Oceanic and Atmospheric Administration, Boulder, CO, USA
[2]School of Chemistry, University of Bristol, Bristol, UK
[3]Hadley Centre, Met Office, Exeter, UK
[4]Climate Science Centre, CSIRO Oceans and Atmosphere, Aspendale, Victoria, Australia
[5]Cooperative Institute for Research in Environmental Sciences, University of Colorado, University of Colorado, USA
[6]Empa, Swiss Federal Laboratories for Materials Science and Technology, Dübendorf, Switzerland
[7]College of Environmental and Resource Sciences, Zhejiang University, China
[8]A/gent b.v. Consultancy, Venlo, Netherlands
[9]Natural Resources Defense Council, USA
[10]Stratospheric Protection Division, Environmental Protection Agency, Washington, DC, USA
[11]Department of Pure and Applied Sciences, University of Urbino, Urbino, Italy
[12]Institute for Atmospheric and Environmental Science, Goethe University Frankfurt, Frankfurt am Main, Germany
[13]Scripps Institution of Oceanography, University of California San Diego, La Jolla, CA, USA
[14]Department of Oceanography, Kyungpook National University, Daegu, Republic of Korea
[15]Center for Global Change Science, Massachusetts Institute of Technology, Cambridge, MA, USA

**Correspondence:** Luke M. Western (luke.western@noaa.gov/luke.western@bristol.ac.uk)

**Abstract.** Global emissions of the ozone depleting gas 1,1-dichloro-1-fluoroethane (HCFC-141b, $CH_3CCl_2F$), derived from measurements of atmospheric mole fractions, have been rising between 2017-2021 despite a fall in reported production and consumption for dispersive uses. This study evaluates the possible drivers behind this renewed rise in emissions. HCFC-141b is a controlled substance under the Montreal Protocol, and its phase-out is currently underway, after a peak in reported

5 consumption and production in developing countries (Article 5) in 2013. If reported production and consumption are correct, it suggests that the 2017-2021 rise is due to an increase in emissions from the bank when HCFC-141b containing appliances reach the end of their life, or from production of HCFC-141b not reported for dispersive uses. Regional emissions have been estimated between 2017-2020 for all regions where measurements have sufficient sensitivity to emissions. This includes the regions of northwestern Europe, east Asia, the USA and Australia, where emissions decreased by a total of 1.6 ± 3.9 Gg

10 yr$^{-1}$, compared to a mean global increase of 3.0 ± 1.2 Gg yr$^{-1}$ over the same period. Collectively these regions only account for around a third of global emissions in 2020. Therefore we are not able to pinpoint the source regions or specific activities responsible for the recent global emission rise.



## 1 Introduction

The global atmosphere has seen a decline in the burden of most ozone-depleting substances since the implementation of the
Montreal Protocol on Substances that Deplete the Ozone Layer (Engel and Rigby, 2019). Under the Protocol's framework,
the global phase-out of production of chlorofluorocarbons (CFCs) and halons for dispersive uses was reportedly completed in
2010. The phase-out (with the exception of very small amounts for the servicing of existing equipment) of their controlled re-
placement gases, primarily hydrochlorofluorocarbons (HCFCs), was completed in 2020 in developed (non-Article 5) countries,
whilst developing countries (Article 5) are in the process of a staged phase-out, to be completed by 2030.

Despite a global ban on CFC production for dispersive uses, recent work found unexpected emissions of trichlorofluo-
romethane (CFC-11) between 2012 and 2017, likely stemming from CFC-11 produced in violation of the Montreal Protocol
after 2010 (Montzka et al., 2018; Rigby et al., 2019; Montzka et al., 2021; Park et al., 2021). These studies provided evi-
dence of renewed dispersive use of CFC-11 in eastern China, which accounted for around 60% of the concurrent increase
in global emissions. Elevated emissions from eastern China of both CFC-12 and $CCl_4$, chemicals involved in the production
of CFC-11, from which they can escape to the atmosphere, suggested that CFC-11 production may have also occurred in this
region. The most likely application of this newly produced CFC-11, which was not reported to the United Nations Environment
Programme's (UNEP) Ozone Secretariat, was as a blowing agent for closed-cell foams (TEAP, 2019).

The most widely used replacement gas for CFC-11 for foam blowing in developing countries was HCFC-141b (1,1-
dichloro-1-fluoroethane, $CH_3CCl_2F$), which also has minor applications as an aerosol, a solvent and feedstock, and is also
an intermediate/by-product during the production of other fluorochemicals. Under the phase-down schedule of the Montreal
Protocol, HCFC-141b should no longer be produced or consumed for dispersive uses in developed countries, and production
should be declining in developing countries since the HCFC phase-out began in 2013. HCFC-141b has a much shorter atmo-
spheric lifetime than CFC-11 (around 9.4 compared to 52 years), and its potential to deplete stratospheric ozone is only around
0.07-0.10 times that of CFC-11 (Burkholder, 2019). Yet, HCFC-141b is still an ozone-depleting substance, with the potential
to delay stratospheric ozone recovery, and, along with other HCFCs, it is also a potent greenhouse gas, with a global warming
potential 800 times that of carbon dioxide over a 100-year time horizon (Burkholder, 2019).

Reported HCFC-141b consumption – defined as production for dispersive uses plus imports minus exports and destruction
– exhibits two peaks (Fig. 1), one in 2002 and one in 2011. In addition to foam blowing, dispersive uses for HCFC-141b are
as an aerosol (1.4 Gg was consumed for aerosols in 2014; 0.7 Gg in 2018) and a solvent (Multilateral Fund, 2019). The use of
HCFC-141b for solvent cleaning has declined, from 4.7 Gg yr$^{-1}$ in 2014 to 3.8 Gg yr$^{-1}$ in 2018, and is predicted to decline
further as more critical uses are given priority (MCTOC, 2018; Multilateral Fund, 2019). HCFC-141b produced for use as a
feedstock is differentiated from dispersive production when reported as its production quantity is not relevant for compliance
with the Montreal Protocol (both production for dispersive uses and feedstock are shown in Fig. 1). Production of HCFC-
141b for feedstock was at a maximum of 18 Gg yr$^{-1}$ in 2011, compared to 118 Gg yr$^{-1}$ in 2011 for dispersive uses, and has
remained at 12-13 Gg yr$^{-1}$ between 2014-2020. However, the proportion of HCFC-141b produced as feedstock increased from
12% in 2014 to 23% in 2020 due to the decline in production for dispersive uses. HCFC-141b can be a feedstock, a by-product

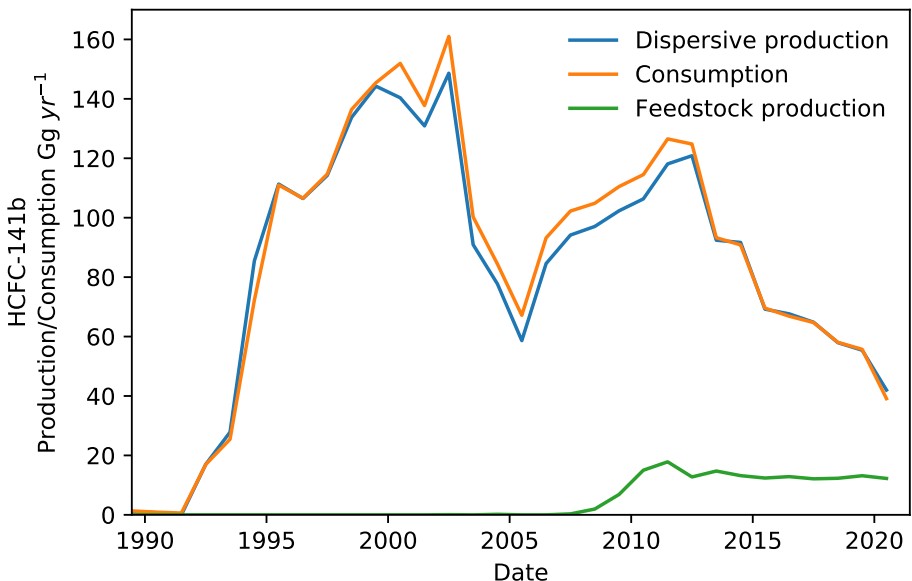

**Figure 1.** Global HCFC-141b production for dispersive uses (blue) and consumption (orange) reported to UNEP. Production of HCFC-141b for use as a feedstock (green) is not included in the reported total for dispersive production or compliance considerations with respect to the Montreal Protocol.

and a target product. Starting out from methyl chloroform or vinylidene chloride (VDC), HCFC-141b, HCFC-142b and HFC-143a are produced (Andersen et al., 2021). HCFC-142b can then be converted to vinylidene fluoride (VDF, HFO-1132a), a refrigerant and also the building block for the fluoropolymer polyvinylidene fluoride (PVDF). Any unwanted HCFC-141b can

50   also be fed into this production chain (TEAP, 2021; Andersen et al., 2021). While the size and exact fate of these production routes are not publicly known, it is possible that the market for PVDF is growing due to its use in Li-ion batteries and other high-tech applications. Previous estimates of global HCFC-141b emissions based on atmospheric observations have generally been consistent with inventory estimates, based on consumption reported to UNEP (neglecting feedstock) and assumptions about rates of release to the atmosphere (Montzka et al., 2015; Simmonds et al., 2017).

55   Regional top-down (based on atmospheric measurements) and bottom-up (based on assumptions about the size and rate of release from various emissive processes, and reported or market-based estimates) HCFC-141b emission estimates are sparse and exist only for northeast Asia, India, western Europe and the USA. Top-down emissions estimates for China through 2017 by Fang et al. (2019b) show emissions declining from $24 \pm 5$ Gg yr$^{-1}$ to $15 \pm 2$ Gg yr$^{-1}$ between 2011 and 2017. A decline in Chinese emissions in recent years is supported using a different set of atmospheric data by the estimates of Yi

60   et al. (2021), albeit with smaller overall emissions, showing Chinese emissions peaking at 16 Gg yr$^{-1}$ in 2014, dropping to 11 Gg yr$^{-1}$ by 2019 (uncertainties were not given). Conversely, bottom-up estimates for China projected a peak in emissions in





2018 (26 Gg yr$^{-1}$) (Wan et al., 2009) or the mid-2020s (31 to 35 Gg yr$^{-1}$) (Wang et al., 2015; Fang et al., 2018), when foam products come to the end of their life following peak consumption. The refrigeration and electric water-heater sectors contribute most significantly to these disposal-related emissions (Wang et al., 2015). Statistics on China's refrigerator production and

disposal projected a continued increase in HCFC-141b emissions from China into 2020, when emissions were estimated to be around 12 Gg yr$^{-1}$ from household refrigerator disposal (Zhao et al., 2011). Top-down emissions estimates for India based on measurements from an aircraft campaign in June-July 2016 were 1.0 (0.7-1.5) Gg yr$^{-1}$ (Say et al., 2019), while estimates for Europe from 2009 are estimated to be in the region of 1.4 (0.8-2.0) Gg yr$^{-1}$ (Keller et al., 2012). Bottom-up estimates for the USA by the U.S. Environmental Protection Agency (EPA) (EPA, 2021) reached a maximum of 9.1 Gg yr$^{-1}$ in 2014,

and declined to 7.1 Gg yr$^{-1}$ in 2017, with a slowing rate of decline to 6.7 Gg yr$^{-1}$ in 2020. The reason for the peak in U.S. emissions following a long-term decline in consumption is likely due to an increase in emissions at the end-of-life of rigid boardstock, commercial refrigeration foams and domestic refrigerator-freezer insulation products.

    This work explores whether an increase in global HCFC-141b emissions, starting in 2018, can be fully attributed to emissions from the HCFC-141b bank due to dispersive production reported to UNEP, or from other activities not that may not be reported

under the Montreal Protocol. Here we present overviews of the data sets and modelling approaches used in Section 2. We present estimates of global HCFC-141b emissions based on atmospheric measurements and reported consumption (Section 3.1) and regional emission estimates for east Asia (Section 3.2), Europe (Section 3.3), the USA (Section 3.4) and Australia (Section 3.5), followed by the conclusions (Section 4).

## 2   Methods

### 2.1   Measurements

We use measurements of dry-air atmospheric mole fractions from two global monitoring networks, the Advanced Global Atmospheric Gases Experiment (AGAGE, Prinn et al., 2018) and the United States National Oceanic and Atmospheric Administration (NOAA) Global Greenhouse Gas Reference Network (Hu et al., 2015, 2016, 2017). Measurements from AGAGE and NOAA stations in the remote atmosphere were used separately to estimate global emissions. Measurements from AGAGE

stations provide regional emissions estimates for Europe, Australia and East Asia, and NOAA measurements for the USA. Figure 2 shows the locations of the measurement stations, and further information is summarised in tables S1 and S2.

    AGAGE HCFC-141b measurement are reported on the Scripps Institution of Oceanography (SIO) 2005 calibration scale. The GCMS-Medusa instruments in Table S1 are cyrogenic pre-concentration systems coupled with gas chromatograph (GC, Agilent) and quadrupole mass selective detector (MSD) (Miller et al., 2008; Arnold et al., 2012). The ADS–GCMS is an

adsorption–desorption system with a gas chromatograph and mass spectrometer (Maione et al., 2013). AGAGE in-situ atmospheric measurements were made with these systems approximately every 2 hours. Paired flask samples were collected at the Taunus observatory (Table S1) and analysed on a GC-quadrupole MSD (Schuck et al., 2018). Before in-situ measurements were available, global emission estimates are based on archived air samples (historic air samples collected and stored). Archived air samples from the Cape Grim Air Archive (CGAA, collected 1978-2009) for the Southern Hemisphere were measured in 2011



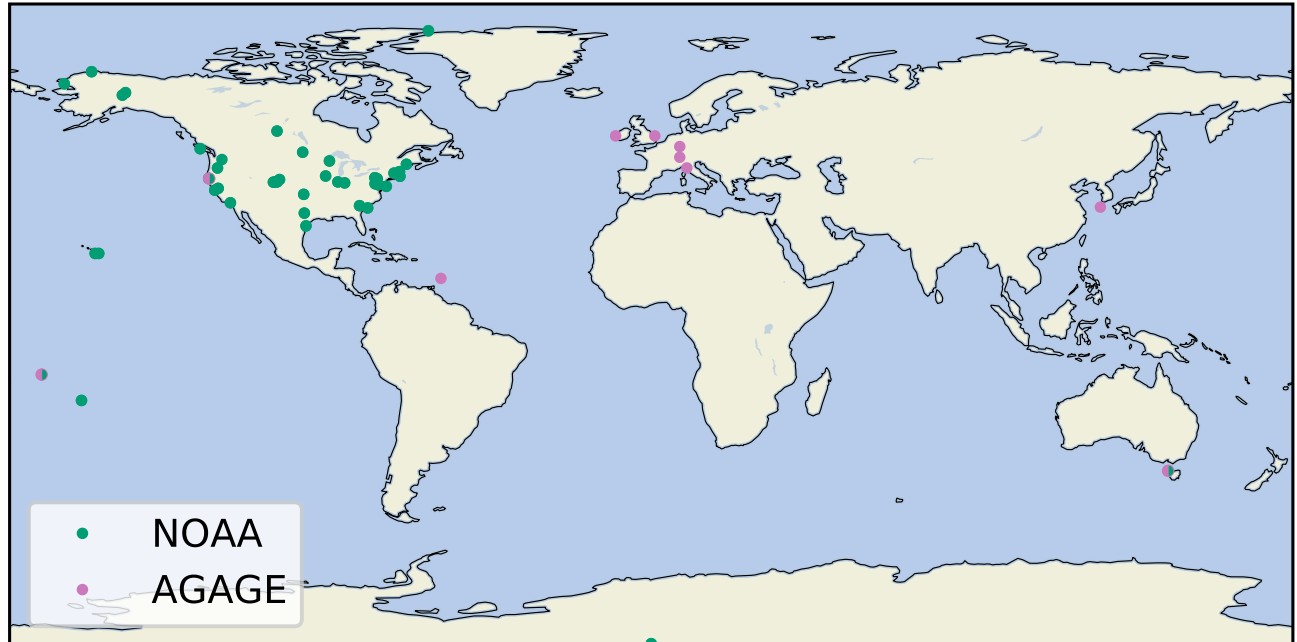

**Figure 2.** Locations of the AGAGE (pink circles) and NOAA (green circles) measurement stations used in this work to estimate global and regional HCFC-141b emissions.

using GCMS-Medusa technology (Fraser et al., 2018). Several CGAA and 126 archived air samples taken at Trinidad Head and other locations in the Northern Hemisphere between 1973–2016 were measured at the Scripps Institution of Oceanography using GCMS-Medusa technology (Mühle et al., 2010; Trudinger et al., 2016; Mühle et al., 2019).

NOAA estimates of the global mean, remote atmospheric abundance of HCFC-141b considered here are derived from measurements of air samples pressurised into paired stainless-steel flasks that are collected approximately weekly at 8 remote

sites (see Table S2). The flasks are shipped to Boulder for analysis on customised GC/MS instruments (Montzka et al., 2015). For deriving continental U.S. emissions from atmospheric measurements, additional flask samples are regularly collected from tall towers (100-400m agl; single flasks per sample, typically) and as profiles from aircraft (6 to 12 flasks collected at regular altitude intervals in a profile up to 8 km asl) at 17 profiling locations (Hu et al., 2017).

## 2.2 Global emission modelling

### 2.2.1 Measurement-based global emissions inference

Global top-down emissions of HCFC-141b are estimated based on atmospheric mole fraction measurements either from the AGAGE or NOAA network, a 12-box model of global transport and inverse modelling. The 12-box model simulates annually repeating advection and diffusion in the global atmosphere (Cunnold et al., 1983; Rigby et al., 2013) and separates the





atmosphere at 30 °N, the equator and 30 °S and at 500 and 200 hPa. The rate of reaction of HCFC-141b with the hydroxyl
radical (OH) was taken from Burkholder (2019), and global mean OH concentrations were inferred in the box model based
on observations of methyl chloroform (Rigby et al., 2013). A first-order stratospheric loss was imposed to give a stratospheric
lifetime of 72.3 years. The overall lifetime of HCFC-141b was 9.3 years in the model.

The measurements from the AGAGE sites that are representative of background conditions in the semi-hemispheres of the
12-box model are used to estimate global emissions (MHD, THD, RPB, SMO, and CGO; see Table S1) after measurements
not representative of background conditions were removed using a statistical algorithm (O'Doherty et al., 2001). When mea-
surements are made in the same latitude band using multiple instruments (MHD and THD), the mean value is used. NOAA
measurements used as input to the 12-box model are shown in Table S2. They were also filtered to select only those thought
to be representative of background conditions, eliminating a small fraction of the entire data record (1.7 % of flask pairs).
Results were also eliminated when measured mole fractions in simultaneously-filled flasks differ by more than 0.28 ppt (3.0
% of all flask pairs). Monthly semi-hemispheric means were derived using a sine-weighting of site latitude. See Tables S1, S2
and Section 2.1 for more details on the measurement sites and instruments used.

Emissions are estimated using an inverse framework (Rigby et al., 2014), through minimisation of a cost function that
constrains the emissions growth rate between years. A priori, the growth rate was assumed to be zero plus or minus 20 % of the
maximum emissions from the EDGAR v4.2 bottom-up dataset (Janssens-Maenhout et al., 2011). Systematic uncertainties in
the inferred emissions, in addition to the measurement error, are derived using a Monte Carlo approach, which includes errors
due to transport, HCFC-141b lifetime (one standard deviation uncertainty of 15 % was assumed, based on (SPARC, 2013))
and instrument calibration. The mole fraction growth rate is calculated as the annual growth rate per month, and is smoothed
using a Kolmogorov–Zurbenko (KZ) filter (Yang and Zurbenko, 2010) using an approximately 18 month window.

### 2.2.2  Consumption-based global emissions modelling

We estimate emissions using reported global consumption data and an adaptation of the methodology employed by Simmonds
et al. (2017), where release rates from the reported bank are estimated to best reproduce the top-down global emissions esti-
mates. We call this approach a top-down-informed bank-model. In a given year, $i$, total emissions to the atmosphere, $E$, are
assumed to come from a combination of prompt releases in the year of consumption, $C$ (due to losses during production and
installation), and emissions from the existing HCFC-141b bank, $B$, following the relationship,

$$E_i = fC_i + gB_i,  \qquad (1)$$

where $f$ and $g$ are the prompt and bank release fractions, respectively, and remain constant over time. The bank grows as,

$$B_i = (1 - f)C_{i-1} + (1 - g)B_{i-1}.  \qquad (2)$$

We extend this approach by using separate consumption data for Article 5 and non-Article 5 countries, which each have their
own prompt and bank release fractions (rather than a single value as used in Simmonds et al. (2017)).

In order to derive values for $f$ and $g$ for both Article 5 and non-Article 5 countries, we use the top-down global emissions
estimates and a statistical framework. For this analysis, we only use emissions estimated using AGAGE data, as these measure-





ments predate the non-negligible global consumption. We constrain $f$ and $g$ using the top-down emissions and the relationship in Equation 1 in a Markov chain Monte Carlo framework, which allows uncertainties to be propagated throughout. No prior constraint is placed on the value of $f$ and $g$, other than that they must be between 0-100 % with equal prior probability. Under

this framework, we simultaneously infer consumption in 2021, which had not been fully reported to UNEP at the time of writing, considering the uncertainties in the top-down emissions and release fractions. For 2021, we assume that all consumption from non-Article 5 countries is less than 1 Gg yr$^{-1}$ (with equal probability for all values between 0-1 Gg yr$^{-1}$) and place no prior constraint on Article 5 consumption, other than it must be a positive value. We make this assumption as, under the phase-out schedule, only very minor consumption (less than 0.5% of baseline usage, which is around 0.2 ODP-Gg for all

HCFCs) would be expected from non-Article 5 countries. Consumption of HCFC-141b in 2020 for non-Article 5 countries was negative, i.e. more HCFC-141b was destroyed than consumed. Therefore, it is reasonable to assume that any consumption, above minor usage, should only occur in Article 5 countries. As such, it is possible to use the estimated 2021 consumption and estimates of release fraction to predict 2021 emissions (with the propagated uncertainty).

## 2.3 Regional modelling

Regional emission estimates for East Asia were derived using four inverse methods: the Bristol-MCMC inversion, Sect. 2.3.1; InTEM, Sect. 2.3.2; EBRIS, Sect. 2.3.3; and FLEXPART-MIT, Sect. 2.3.4. Regional emissions for Northwest-Europe (NW-Europe) and Australia were derived using InTEM, Sect. 2.3.2. Regional emissions for the contiguous United States were derived using the NOAA framework, Sect. 2.3.5. Inverse methods were run with independent choices to a priori emissions, statistical models, tranport set-ups and treatment of measurement data sets, in instances where multiple estimates were performed

for the same region.

### 2.3.1 Bristol-MCMC

Linear sensitivities of measured mole fraction to emissions (or 'footprints') were derived using the UK Met Office NAME model (Jones et al., 2007). Sensitivities were calculated for a computational domain bounded at 5°S and 74°N and 55°E and 192°E. Meteorology from the UK Met Office Unified Model (Met-UM Global, Walters et al., 2014) drive the transport,

which increases in resolution from 0.563° to 0.141°longitude and 0.375° to 0.094°latitude between 2008-2020. The temporal resolution remained at 3 hours throughout this period. Around 20,000 particles are released each hour within the NAME model domain, and a measurement is deemed to be sensitive to emissions when a particle is transported within the lowest 40 m above ground level of the model domain. Sensitivities are output on a grid of 0.234°longitude by 0.352°latitude.

Emissions are estimated using a Bayesian Markov chain Monte Carlo inverse framework (Ganesan et al., 2014; Say et al.,

2019) independently each year by scaling an a priori emissions field and the mole fraction contribution from the model boundary using measurements from Gosan station averaged into 12-hourly bins. The a priori emissions field is 1.16 Gg yr$^{-1}$ for eastern China, 0.13 Gg yr$^{-1}$ for South Korea, 0.17 Gg yr$^{-1}$ for western Japan and 0.17 Gg yr$^{-1}$ for North Korea. Emissions were distributed equally in space over land with a log-normal uncertainty, where the distribution is described by the shape parameters $\mu$, or log-median value, equal to 0.2, and $\sigma$, equal to 0.8. The a priori mole fraction at the model boundary was





taken from the AGAGE 12-box model (Rigby et al., 2014). These are assigned a prior log-normal distribution, with a prior
distribution with $\mu$ equal to 0.004 and $\sigma$ equal to 0.02. In addition to the measurement error, which was assumed prior to
inference, we estimate the model transport error in a normal likelihood, assigning it a log-normal prior distribution of $\mu$ equal
to 0.2 and $\sigma$ equal to 0.8. The computational domain was divided into 151 basis functions using a quadtree algorithm, and the
mole fractions at the boundaries were estimated in each cardinal direction (Say et al., 2019; Western et al., 2021). We use a

No-U-Turn (NUTS) sampler to sample the emissions and boundary influence and a Slice sampler to sample the model error
(Salvatier et al., 2016) using 90,000 sampling steps (with an addition 10,000 discarded at the beginning of the sampling chain).
Convergence was checked using a Gelman-Rubin diagnostic (Gelman and Rubin, 1992) on multiple chains.

### 2.3.2   InTEM

The InTEM inversion methodology is described in Manning et al. (2021). Briefly, the footprint sensitivities were generated

using NAME as described in Sec. 2.3.1 using global UM data. Emission estimates for East Asia were derived using measurement data from Gosan, averaged into four-hourly time intervals. Prior mean emissions in East Asia were uniformly distributed
over all land areas within the computational domain, with total emissions equal to 50 Gg yr$^{-1}$, and a one standard deviation
uncertainty equal to of 300% of the prior mean emissions. This resulted in the following prior emissions: eastern China 2.6 $\pm$
24.3 Gg yr$^{-1}$; South Korea 0.3 $\pm$ 8.7 Gg yr$^{-1}$; western Japan 0.4 $\pm$ 9.9 Gg yr$^{-1}$; and North Korea 0.4 $\pm$ 9.0 Gg yr$^{-1}$.

Emission estimates for Europe used measurement data from Mace Head, Jungfraujoch, Monte Cimone, Tacolneston and
Taunus. For Europe, the footprints were bounded by a computational domain of 98.1°W to 39.6°E, 10.6°N to 79.2°N, using
global UM data nested with higher resolution Unified Model meteorology over the United Kingdom and Ireland (UK-V, Tang
et al., 2013). The prior emissions for countries in NW-Europe were: Belgium and Luxembourg 0.1 $\pm$ 4.4 Gg yr$^{-1}$; France 1.8
$\pm$ 18.1 Gg yr$^{-1}$; Germany 1.1 $\pm$ 13.5 Gg yr$^{-1}$; Ireland 0.3 $\pm$ 6.8 Gg yr$^{-1}$; Netherlands 0.1 $\pm$ 4.8 Gg yr$^{-1}$; and the UK 1.1 $\pm$

12.7 Gg yr$^{-1}$.

   Emissions estimates for Victoria, Tasmania, southern and south-western New South Wales and eastern South Australia are
based on measurements at Cape Grim, averaged every 4 hours. They are then scaled by population (a factor of 2.6) to the whole
of Australia. Footprints were bound to a computational domain of 70.0°E to 214.7°E and 65.0°S to 5.0°N. The prior emissions
for Australia were 0.4 $\pm$ 2.0 Gg yr$^{-1}$, distributed by population density.

Prior boundary mole fractions at each station were estimated using a fourth order polynomial fitted to measurements that
were representative of background air, having little influence from populated areas, and refined within the InTEM framework
(see Manning et al., 2021).

### 2.3.3   EBRIS

A detailed description of the Empa Bayesian Regional Inversion System (EBRIS) is given in Henne et al. (2016). The method

used measurements from the Gosan station, averaged every 3 hours, to derive emissions. Footprint sensitivities for East Asia
were derived using the FLEXPART transport model (Pisso et al., 2019), which was driven by operational ECMWF analysis
meteorology with 1°$\times$ 1° resolution, reducing to 0.2°$\times$ 0.2° resolution for northeastern China (105°E to 125°E and 30°N to



50°N). In each three-hour interval, 50,000 particles were released and tracked backward for 10 days. Footprints were derived for a large northern-hemispheric domain at a resolution of $0.125° \times 0.125°$ and a particle sampling height of 100 m.

Inversions were carried out independently for average annual emissions. The inversion grid (state vector) was limited to the domain 80°E to 140°E and 20°N to 60°N. Grid resolution was inversely proportional to the average footprint, with smaller grid cells ($0.125° \times 0.125°$) close to the measurement site and large grid cells ($8° \times 8°$) away from the measurement site. Approximately 500 grid cells were included in the inversion grid, depending on the data coverage of individual years.

     Baseline concentrations were estimated using the Robust Estimation of Baseline Signal (REBS) method (Ruckstuhl et al., 215   2012) applied to the observations at Gosan. Bi-weekly baseline concentrations were part of the state vector and were optimised during the inversion step.

     The same a priori emissions were assigned for each year in the inversion. Homogeneous a priori distributions were prescribed in each of the seven focus regions (Western China 6.3 Gg yr$^{-1}$, Eastern China 6.7 Gg yr$^{-1}$, North Korea 0.1 Gg yr$^{-1}$, South Korea 1.0 Gg yr$^{-1}$, Western Japan 2.0 Gg yr$^{-1}$, Eastern Japan 0.6 Gg yr$^{-1}$, Taiwan 0.2 Gg yr$^{-1}$).

The a priori covariance and data-mismatch covariance were estimated using a log-likelihood optimisation of parameters describing the covariance (Henne et al., 2016). As part of this optimisation the domain total a priori uncertainty was determined to 140 to 160 % varying from year to year.

### 2.3.4   FLEXPART-MIT

The FLEXPART-MIT inversion is described in Fang et al. (2019a). The FLEXPART-MIT inversion also used FLEXPART 225   to derive footprint sensitivities, but under a different setup to Sect. 2.3.3. In every three hour interval, 40,000 particles were released and tracked backwards for 20 days. Meteorology was driven by operational ECMWF analysis at $1° \times 1°$ global resolution over a global computational domain.

     A priori flux fields were spatially uniform over continental eastern Asia, with no emissions from the ocean. Emissions were estimated using a variable-resolution grid. The grid was finest ($1° \times 1°$) in eastern China and other eastern Asian countries, 230   and a coarser grid resolution ($24° \times 24°$) was used outside this area. A priori emissions estimates of HCFC-141b were 14.5 Gg yr$^{-1}$ for China (1.5 Gg yr$^{-1}$ for eastern China) and 1.2, 0.27 and 0.27 Gg yr$^{-1}$ for Japan, South Korea and North Korea, respectively. Prior uncertainty was arbitrarily set to 1,000% of the a priori estimate, which assumed a spatial correlation length of 300 km. The background mole fractions was estimated in 7-day periods. Model-measurement uncertainty for each 24-h-averaged observation was estimated using the quadratic sum of: 1% of the baseline value (as a measure of baseline uncertainty), 235   the measurement repeatability, and the standard deviation of the 24-h variability (as a measure of the model–data 'mismatch' uncertainty).

     The inverse framework utilises a Bayesian framework using an analytical solution to a normal likelihood and prior (Stohl et al., 2009) for each year.





### 2.3.5 NOAA

The NOAA inversion framework was first presented in Hu et al. (2015) and is used to derive US emissions for a number of ozone depleting substances and their substitutes (Hu et al., 2016, 2017). A similar methodology to that used previously was used to derive US emissions of HCFC-141b for 2015 – 2020. A total of six ensemble inversions with identical a priori emission fields were conducted for deriving US HCFC-141b emissions (three approaches to estimate background mole fraction using two transport models). The US a priori HCFC-141b emissions were 4.1 Gg yr$^{-1}$ and were scaled by population density to

generate $1° \times 1°$ a priori emissions. Prior uncertainties were estimated by maximum likelihood estimation (Michalak et al., 2005; Hu et al., 2015). Footprints were simulated by two transport models, the Hybrid Single-Particle Lagrangian Integrated Trajectory Model (HYSPLIT) for 2015 – 2020 and the Stochastic Time-Inverted Lagrangian Transport (STILT) model for 2015 – 2017. The HYSPLIT model was run with 500 particles back in time for 10 days and driven by the North American Mesoscale Forecast System (NAMS) with 40 sigma-pressure levels and 12-km horizontal resolution over the contiguous US. The NAMS

meteorology was nested with a global meteorological field, the US National Centers for Environmental Prediction (NCEP) 0.5° Global Data Assimilation System (GDAS0.5) with 55 sigma-pressure levels (before June 2019) and the NCEP 0.25° Global Forecast System (GFS0.25) forecast model with 55 sigma-pressure levels (after June 2019). The STILT simulation was also run with 500 particles back in time for 10 days. It was driven by the Weather Research and Forecasting Model (WRF) with 10-km resolution over North America and 40-km resolution outside of North America.

Three different approaches were used to derive background HCFC-141b mole fractions for measurements made in the US (see details described in Hu et al., 2017, 2021). All three approaches were first based on a 3D background field (as a function of time, latitude, and altitude) constructed from atmospheric observations far away from emission sources, i.e. those made over the Pacific and Atlantic Ocean basins near Earth's surface and in the free troposphere above North America. In the first approach, we estimated the background mole fraction associated with each measurement made in the USA by assigning mole

fractions from this 3D background field based on the sampling time, latitude, and altitude for measurements made in the USA. In our second approach, we considered air back-trajectories for individual measurements made in the USA. We estimated the time and location of each particle exiting the planetary boundary layer of the contiguous US based on its back-trajectories and assigned the mole fraction from the 3D background at the exiting time and location. The third background estimate for each measurement in the US is an average of 500 background estimates based on the 500 back-trajectories. In this third approach,

we considered possible biases in transport simulations and lack of mole fraction information in the planetary boundary layer of the US, because some particles remained in the planetary boundary layer of the US after 10 days of running the transport model backward in time. Thus, we applied a likely bias correction to the background derived from the second approach by comparing observed mole fractions with estimated background mole fractions for a subset of observations with minimal surface emission influence, i.e. summed footprint over populated areas (areas with more than 10 persons km$^{-2}$) less than 0.1 ppt (pmol m$^{-2}$

s$^{-1}$)$^{-1}$. The final annual US emissions were reported as the ensemble mean and uncertainties from the 6 inversions. The



reported uncertainties are a one-sigma uncertainty ($\sigma_t$) calculated as

$$\sigma_t = \frac{1}{6}\sum_{i=1}^{6}\sigma_i + \sigma_s \tag{3}$$

where $\sigma_i$ is the one-sigma posterior uncertainty derived from each inversion; $\sigma_s$ denotes the one-sigma spread of the posterior emissions derived from all six inversions.

## 3 Results and discussion

### 3.1 Global mole fraction and emissions

As expected from the Montreal Protocol mandated HCFC phase-out schedule, global emissions of HCFC-141b had been declining since 2012 (Montzka et al., 2015; Simmonds et al., 2017; Engel and Rigby, 2019). However, since 2017, an increase in emissions is evident from recent changes in the distribution and global mean mole fraction of HCFC-141b. The growth rate of the global mole fractions had continuously slowed since 2012, and the global mole fractions started to decline during 2017 (Figs. 3(a) and (b) for AGAGE and NOAA, respectively). Such a slow-down and decline is expected as emissions decrease and HCFC-141b present in the atmosphere is destroyed by reactions with the OH radical. Since 2018, however, the rate of decline in global mole fraction slowed and, as of 2019, annual mean growth rates are positive again. This increase is likely being driven by increased emissions in the northern hemisphere, as the increase in the growth rate in the northern hemisphere leads that in the southern hemisphere in recent years, just as it did in earlier periods when the global HCFC-141b growth rate and emissions were rapidly increasing (e.g., 1992-1996 and 2009-2012). As of 2019 there is a growing difference between the observed mole fraction in the northern and southern hemispheres (mean difference of 1.8 ppt in 2018 and 2.2 ppt in 2021).

Using AGAGE and NOAA measurements, we show that emissions of HCFC-141b started to increase again in 2018, despite the reported reduction in global dispersive production and consumption expected from the Montreal Protocol control schedules (Figs. 1 and 4(a)). Aggregated emissions from 2017 to 2021 have increased by an additional $4.4 \pm 1.9$ Gg (AGAGE) or $7.5 \pm 0.5$ Gg (NOAA), an average linear increase at a rate of $1.1 \pm 0.2$ Gg yr$^{-1}$ per year (AGAGE) or $1.6 \pm 0.1$ Gg yr$^{-1}$ per year (NOAA). This assumes that all uncertainty is only due to the random error component of the emissions estimates (i.e. systematic uncertainty causes an offset in emissions and does not impact year-to-year variability). Note that Figure 4 shows the uncertainty due to the combined random and systematic error components.

To understand this reversal, we estimate global emissions of HCFC-141b using reported consumption data by adapting the approach taken by Simmonds et al. (2017), as detailed in Section 2.2.2. We estimate the prompt release fraction to be 24 (17-31) % and 37 (36-39) % for Article 5 and non-Article 5 countries, respectively (68 % uncertainty interval). Our estimated bank release fraction is 3.6 (3.0-4.2) % and 2.8 (1.9-3.9) % for Article 5 and non-Article 5 countries, respectively, and is assumed to be unchanged since 1989. Simmonds et al. (2017) used an aggregated HCFC-141b prompt release fraction of $34.5 \pm 4.0$ %, along with a bank release fraction of 2.2 %. Our estimated released fractions thus broadly agree with that used by Simmonds et al. (2017), and that previously estimated for closed-foams containing CFC-11 (TEAP, 2019), which is 25-35 % for the





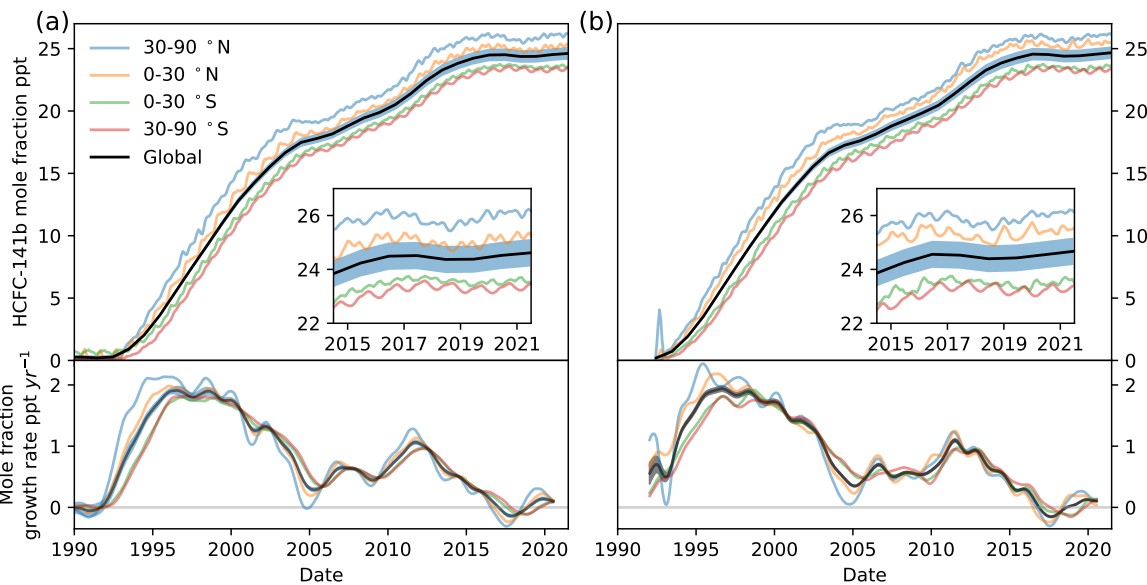

**Figure 3.** HCFC-141b global and semihemispheric mole fraction (top) and growth rate (bottom) derived from AGAGE (a) and NOAA (b) measurements.

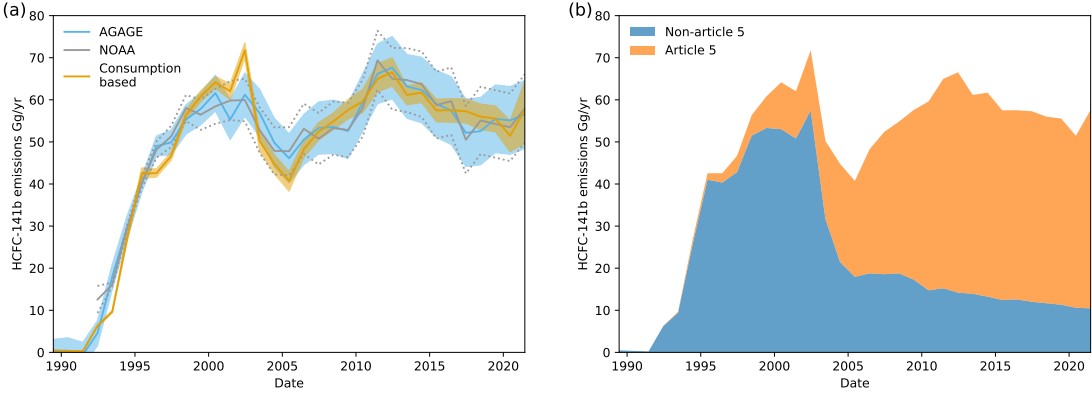

**Figure 4.** (a) HCFC-141b global emissions derived from AGAGE and NOAA measurements, and reported consumption data and estimated emission release fractions, where consumption has been predicted for 2021, assuming consumption in non-Article 5 countries is less than 1 Gg yr$^{-1}$ and no constraints on Article 5 consumption. (b) a breakdown of HCFC-141b global emissions from Article 5 and non-Article 5 countries using the consumption-based emissions estimate, where 2021 consumption has been estimated from the top-down emissions data.





prompt release fraction and 1.5-4.2 % for the bank release fraction. Using the reported consumption and estimated release fraction, we derive the respective emissions using the top-down-informed bank-model. Under the constraints of the top-down-informed bank-model, we are able to estimate the necessary consumption needed to produce the observed top-down emissions in 2021, which has not yet been fully reported. This uses the reported consumption until 2020 and the estimated emissions rate from the bank to estimate the needed 2021 consumption to reproduce the top-down emissions estimate. This analysis, under the constrains of the model, suggests that consumption has likely risen between 2020 and 2021, from 39 Gg yr$^{-1}$ in 2020 to 67 (28-95) Gg yr$^{-1}$ in 2021. This consumption estimate for 2021 is very uncertain due to the uncertainty in the top-down derived emissions, and so a decline in global emissions, and thus decline in consumption, cannot be excluded. Increasing consumption between 2017-2020 is not possible in this model, due the constraints imposed by the reported production and bank behaviour. An alternative conclusion that could be drawn from this analysis, under the assumption that reported consumption should decrease in 2021, is that the emissions rate from the bank has changed in more recent years, meaning that the inferred emissions rate, which is constant in time, is no longer representative and cannot adequately model emissions.

Figure 4(a) shows a comparison of the top-down, atmospheric measurement-derived emissions, using the 12-box model and without using information about reported consumption, and the emissions estimated using top-down-informed bank-model. Although these estimates are unique, they are not independent due to both top-down estimates using the same methodology to derive emissions using AGAGE and NOAA measurements and the dependence of the top-down-informed bank-model on the estimates using AGAGE measurements. Figure 4(b) shows a breakdown of the mean emissions from non-Article 5 and Article 5 countries under the top-down-informed bank-model using reported consumption data and assumptions inherent in the model, where 2021 consumption has been estimated. This imposes that all consumption in 2021 (over a maximum of 1 Gg yr$^{-1}$) must come from Article 5 countries (see Sect. 2.2.2).

The timing of this increase in HCFC-141b emissions is nearly coincident with the rapid post-2018 global decline in CFC-11 emissions (Montzka et al., 2021), which may suggest that the enhanced apparent use and production of CFC-11 during 2012-2018, which was likely used for foam production, may have transitioned to HCFC-141b. An explanation for the recent rise in global HCFC-141b emissions is, however, further complicated by a lack of understanding of the time-dependent changes in emissions from the existing HCFC-141b bank. Wang et al. (2015) suggest that HCFC-141b emissions in China are expected to peak in 2025-2027, 15 years after peak consumption in China. This is attributed to the release of HCFC-141b from foams during the disposal of insulated refrigerators and electric water-heaters at their end of life around 15 years after peak consumption. If such a pattern predicted in China is applied to all developing countries, peak emissions from all developing countries would be expected around 2026-2027, 15 years after their peak consumption in 2011-2012, and could perhaps explain the observed emissions increase. However, a study in Lahore, Pakistan showed that HCFC-141b emissions due to waste and disposal of HCFC-141b containing refrigerators fell between 2005 and 2013 (Ul-Haq et al., 2016). It may therefore not be appropriate to extrapolate the bottom-up predicted bank emission trends in China to other developing countries.

The top-down-informed bank-model neglects emissions from feedstock and other so-called non-dispersive production. Typical emissions factors associated with fluorochemical production is around 4% (IPCC, 2019). If applied to global feedstock production, additional feedstock-related emissions would be 0.5 Gg yr$^{-1}$ between 2017-2020, not enough to explain the global





increase, with no increasing trend. Emissions factors from individual production facilities may vary considerably (0.1-20%, 95% uncertainty, IPCC, 2019). A universal transition in leakage from 0.1% of feedstock production in 2017 to 20% in 2020 across all production facilities would result in an additional 3.0 Gg yr$^{-1}$ of emissions, compared to a mean global increase

in the NOAA and AGAGE estimates of 3.0 ± 1.2 yr$^{-1}$ during the same period. A sustained rapid deterioration in feedstock production losses since 2017 seems an unlikely scenario to explain the global increase in emissions.

In the following section we turn to regional emissions estimates of HCFC-141b, where observations are available, to further diagnose the driver behind the global rise.

## 3.2 East Asian emissions

Atmospheric measurement-based emissions estimates for East Asia are based on four different inverse methods using measurements from Gosan station, South Korea (see section 2.1) and are combined into a single estimate using Monte Carlo sampling. These results provide emissions estimates for South Korea, North Korea and two regions that we denote eastern China (which includes the provinces of Anhui, Beijing, Hebei, Jiangsu, Liaoning, Shandong, Shanghai, Tianjin and Zhejiang) and western Japan (which includes the regions Chūgoku, Kansai, Kyūshū and Okinawa, and Shikoku), as measurements at Gosan have little

sensitivity to emissions in western China or eastern Japan.

Emissions of HCFC-141b from eastern China show an increase in emissions from 3.7 (1.8-4.8) Gg yr$^{-1}$ in 2008 to 7.7 (6.4-10.0) Gg yr$^{-1}$ in 2020. The mean emissions of the inverse frameworks (Figure 5) generally agree well with bottom-up inventory estimates for China (Fang et al., 2018) scaled down to the eastern China region by population or gross domestic product from emissions from the whole of China (the range is shown by the orange shading in Figure 5). Given the uncertainties in the

top-down modelling and the uncertainties likely present in the bottom-up model (which were not given in the study). If the bottom-up predicted emissions come to fruition, HCFC-141b emissions from China will continue to rise until the mid-2020s. Our results presented here for eastern China contrast with top-down estimates presented for the whole of China in previous studies (Fang et al., 2019b; Yi et al., 2021), which suggest a continual fall in emissions.

Emissions from South Korea (around 1-3 Gg yr$^{-1}$) are smaller than from eastern China and it is uncertain whether they are

increasing or decreasing. Emissions from western Japan and North Korea are smaller (generally less than around 1 Gg yr$^{-1}$). Japan, in contrast to other East Asian countries, legislates the removal and destruction of HCFC refrigerants and HCFCs in foams within appliances through the Act on Rational Use and Proper Management of Fluorocarbons.

## 3.3 Northwestern Europe emissions

Emissions for Northwestern Europe (Belgium, France, Germany, Ireland, Luxembourg, the Netherlands and the United King-

dom) were estimated using the InTEM inversion system, based on measurements from Mace Head in Ireland, starting in 1994, with additional measurements from Jungfraujoch in Switzerland from 2008, Monte Cimone in Italy from 2012, Tacolneston in the UK from 2012 and Taunus in Germany from 2013. NW-Europe sees a sharp fall in emissions after 2003, Figure 6(a), from 3.6 ± 0.9 Gg yr$^{-1}$ in 2003 to 2.2 ± 1.0 Gg yr$^{-1}$ in 2004 and then 1.3 ± 0.9 Gg yr$^{-1}$ in 2005, with timing consistent with the





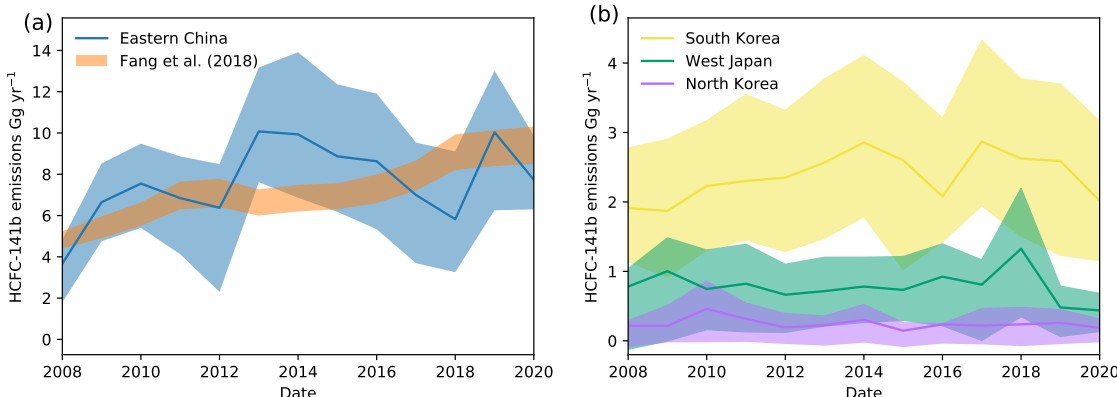

**Figure 5.** Emissions estimates for East Asia. (a) Combined top-down emissions from four inversion frameworks (two in 2020) for eastern China with their 68% uncertainties (blue). Bottom-up emissions estimates for eastern China (orange) estimated by scaling down the Fang et al. (2018) estimate for the whole of China by either population or gross domestic product to eastern China. The shading shows the range between these two metrics for scaling. (b) Top-down estimates for South Korea, western Japan and North Korea.

phase-down of HCFC production in those countries. Emissions have steadily fallen since, reaching $0.4 \pm 0.1$ Gg yr$^{-1}$ in 2020.
Emissions from individual countries are presented in the supplementary information.

Emissions in NW-Europe peak coincidentally with high consumption in developed countries. This contrasts with the suggestion of an emissions peak around 15 years after peak consumption in the bottom-up projections for China. However, lifetimes of products and the disposal methods employed likely differ between these developed and developing countries. The European Union's Regulation (EC) 1005/2009 (and similar legislation in the UK) requires that appliance foam insulation controlled un-
der the Montreal Protocol is recovered at disposal to be recycled or properly destroyed. Therefore we expect relatively little end-of-life emissions in NW-Europe and our results provide evidence that such legislation has, likely, been effective.

### 3.4 Contiguous USA emissions

Top-down emissions estimates for the contiguous USA from 2015 to 2020 (Fig. 6(a)) derived from measurements made by NOAA's North American sampling network (see Sections 2.3.5 and 2.1) using two atmospheric transport models and the
NOAA inversion framework (Section 2.3.5, Hu et al., 2016). Taking an average of the two top-down estimates for the USA, emissions likely increased slightly between 2015 and 2017, with $5.5 \pm 0.5$ Gg yr$^{-1}$ in 2015 and $6.4 \pm 0.5$ Gg yr$^{-1}$ in 2017. The HYPLIT-NAM model estimate suggests that emissions remain unchanged in 2018 at $6.3 \pm 0.4$ Gg yr$^{-1}$ and fall to $5.3 \pm 0.7$ Gg yr$^{-1}$ in 2020. Bottom-up emissions estimates by the U.S. Environmental Protection Agency (EPA, 2021) over the same period (Fig. 6(a)) suggest that emissions fell from 8.6 Gg yr$^{-1}$ in 2015 to 7.1 Gg yr$^{-1}$ in 2017, to 6.7 Gg yr$^{-1}$ in 2020.
The discrepancy in the trend between the top-down and bottom-up estimates in 2015-2016 is unclear, however both show



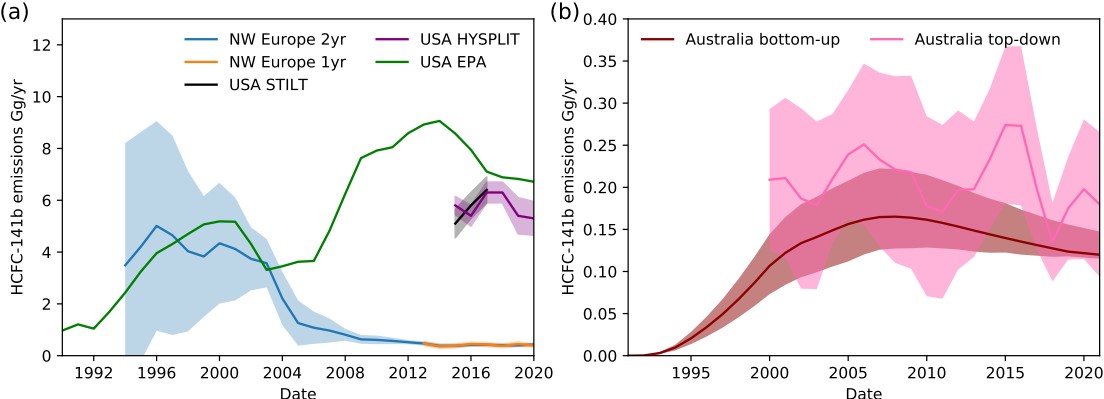

**Figure 6.** (a) Annual InTEM emissions from Northwestern Europe using a two year inversion period (blue) or only a single year (orange), and from the contiguous USA using two transport models, HYSPLIT (purple, 2015-2020) and STILT (black, 2015-2017). Bottom-up estimates for the USA are shown in green. (b) Annual InTEM emissions for Australia (pink) and bottom-up emissions estimated using consumption data (dark red).

only small changes emissions between 2017-2020. The NOAA top-down estimates are 10-40 % smaller than the U.S. EPA bottom-up estimates (assuming that the non-contiguous states and territories are not responsible for the discrepancy).

Consumption of HCFCs in the USA from 2002 to 2003 fell to around half and has continued to decline. The U.S. EPA emission estimate suggests peak emissions many years after a peak consumption, different from what the NW-European top-down emissions show. The reason for a slowing in the decline of emissions since 2017 is not fully clear, but the bottom-up model suggests that it may be due to post-disposal emissions from polyurethane foams used for domestic refrigerator and freezer insulation, occurring during the 26 years following disposal (EPA, 2021). This assumes that domestic refrigerators and freezers have a lifetime of 14 years and that the leakage post-disposal (2 %) is higher than that during the appliance lifetime (0.25 %). In the USA, a gradual increase in HCFC-141b emissions due to the disposal of domestic refrigerants following a peak in consumption may be occurring, much like that predicted to occur in China, offsetting reductions in other banks. Unlike the European Union, which mandates destruction of HCFC-141b within refrigerator foams, the U.S. EPA's Responsible Appliance Disposal (RAD) program is voluntary. This may explain different U.S. and European HCFC-141b emission patterns, despite similar consumption patterns.

### 3.5 Australian emissions

Top-down HCFC-141b emissions estimates using InTEM show that emissions in Australia have remained largely constant from 2000 to 2021, perhaps showing a slight decline (Fig. 6(b)).

Australia has not produced or exported HCFC-141b and therefore HCFC-141b consumption is governed by imports, assuming that no significantly stockpiling for later use occurred. Australian HCFC-141b imports/consumption commenced in the





early-1990s, reaching 0.8-0.9 Gg in 1999, and then declining to zero by 2012, well ahead of the mandated HCFC phase-out
schedule (Dunse et al., 2021). From 1991 to 2012, Australian HCFC-141b imports totalled 7.8 Gg.

Assuming that emissions have therefore only come from the bank, and using the emissions factor derived for non-Article 5 countries from section 3.1, we derive a bottom-up emissions estimate based on consumption in Australia. The bottom-up model likely underestimates the emissions release rate because prompt emissions, which we assume to include emissions during installation and immediate usage, have not been included. The top-down and bottom-up estimates for Australia generally
agree, given the large uncertainties. Australia has no laws to mandate the destruction of HCFCs within appliance insulating foams at disposal. However, there is no obvious increase in HCFC-141b emissions in Australia in recent years due to an increase in the rate of emissions from foam containing appliances when they reach their end of life (within uncertainties).

## 4 Conclusions

For five years after the reported 2012 peak in HCFC-141b production, global emissions of the ozone depleting substance
HCFC-141b declined. However, the trend reversed and emissions have increased by $3.0 \pm 1.2$ yr$^{-1}$ or 6% through 2021 compared to 2017, even though reported production for dispersive uses continued to decrease. This timing is similar to a decline in global emissions of CFC-11, the ozone depleting substance replaced by HCFC-141b for foam blowing, following a period of increasing global CFC-11 emissions. Due to a current incomplete understanding of the size and behaviour of the global HCFC-141b bank, it is uncertain whether the emission rise is due to production not reported for dispersive uses, as
suggested by a simple bottom-up model constrained by measurement-derived emissions, due to emissions at the end-of-life of foam-containing products, or a combination thereof.

An increase due to emissions from the bank is suggested by bottom-up emissions projections for China due to the disposal of HCFC-141b foam containing appliances. However, according to bottom-up and top-down estimates, emissions from China alone cannot explain the global rise. This pattern of bank-related emissions years after peak consumption may not apply to other
developing countries. Emissions in the USA and Australia, where there are no strict requirements in place for the destruction of HCFCs upon appliance disposal, do not show evidence of declining, or increasing, emissions in recent years. This contrasts with the European Union, which mandates more complete HCFC destruction upon disposal, where emissions have decreased continuously since a phase-out of consumption began. A lack of declining emissions in the USA and Australia may support the argument that a change in emissions rates from the foam banks after the disposal of foam-containing appliances are driving
the rise in emissions.

Emissions of HCFC-141b feedstock is unlikely to be the cause of the global increase, unless emission rates during production have rapidly increased, from near-zero losses to around a fifth of production. The amount and fate of HCFC-141b produced as a byproduct is unknown, and we have no evidence to suggest that this pathway leads to significant emissions or would have changed substantially since 2017 in a way that could explain the global emission increase.
The regional-scale emissions considered here only account for around one third of global emissions in 2020. The combined regional-scale emissions decreased by $1.6 \pm 3.9$ Gg yr$^{-1}$ between 2017-2020, compared to a mean global increase in the



NOAA and AGAGE estimates of $3.0 \pm 1.2$ Gg yr$^{-1}$ during the same period. It seems likely that a substantial recent increase in emissions is coming from regions that we have not studied here.

We cannot demonstrate a conclusive driver behind the 2017-2021 increase in global emissions, given the information available. A better understanding of the behaviour of the HCFC-141b bank and its expected emissions, and more widespread measurement-based emissions monitoring, would aid in understanding the causes for the current rise in HCFC-141b emissions.

*Code and data availability.* AGAGE data are available at http://agage.mit.edu/data/agage-data (last access: 17 March 2022) and https://doi.org/10.15485/1841748 (Prinn et al., 2022). NOAA atmospheric observations are available at the NOAA/GML website (https://gml.noaa.gov/hats/, last access: 6 April 2022). The 12-box model is and the inverse method used to quantify emissions are available via Github, https://github.com/mrghg/py12box and https://github.com/mrghg/py12box_invert. The NAME model and InTEM models are available for research purposed via request to the UK Met Office (enquiries@metoffice.gov.uk) or on request to AJM or ALR. The FLEXPART model is available from https://www.flexpart.eu. The EBRIS algorithm is available from https://doi.org/10.5281/zenodo.1194642 (Henne, 2018). The Bristol-MCMC inversion code is available at https://github.com/ACRG-Bristol/acrg. The code for the FLEXPART-MIT inversion is available on request from XF. The code for the NOAA inversion is available on request from LH.

*Author contributions.* LMW lead the writing of the manuscript with contribution from all coauthors. LWM, ALR, AJM, CMT, LH, SH and XF performed inverse modelling. LMJK provided data on reported consumption and production. DSG provided bottom-up U.S. HCFC-141b emissions. LMJK, CT and DSG provided input on HCFC-141b uses, the behaviour of the banks and appliance lifecycles. BD provided data and analysis on Australian consumption and emissions. JA, AE, PF, CH, PBK, MM, SAM, JM, SO, HP, SP, SR, PKS, DS, RS, TS, CS, KMS, IV, MKV and DY provided measurement data. SAM, RGP, MR and RFW provided oversight to the work and the measurement networks.

*Competing interests.* We declare no competing interests.

*Acknowledgements.* We thank the site operators for their continued support to maintain the measurements at the AGAGE and NOAA stations, and the technical and logistical support of NOAA and CIRES personnel facilitating the collection of tower and aircraft samples throughout North America. We thank Arlyn Andrews for providing the WRF-STILT footprints, Nada Derek for Cape Grim data analysis. We greatly thank Phil DeCola for supporting some of NOAA's inverse modeling analyses. The NASA Upper Atmosphere Research Program supports AGAGE (including partial support of Mace Head, Ragged Point and Cape Grim and full support of Trinidad Head and Cape Matatula) through grant NNX16AC98G to MIT, and grants NNX16AC96G and NNX16AC97G to SIO and multiple preceding grants. Mace Head and Tacolneston station and InTEM are supported by the UK Department of Business, Energy and Industrial Strategy (BEIS contract 1537/06/2018). Cape Grim station is supported by the Australian Bureau of Meteorology, CSIRO, the Australian Department of Agriculture, Water and the Environment (DAWE) and Refrigerant Reclaim Australia (RRA). Funding for NOAA measurements was provided in part by



the NOAA Climate Program Office's Atmospheric Chemistry, Carbon Cycle, and Climate (AC4) program, and by the NOAA Cooperative Agreement with CIRES, NA17OAR4320101. NOAA's HYSPLIT-NAMS footprints simulations were supported by NOAA Climate Program Office's AC4 program and Climate Observations and Monitoring (COM) program, grant number NA21OAR4310233. NOAA's inverse modeling analysis was partially supported by the Grantham foundation and the Gist.Earth LLC. Computation at the University of Bristol was 
470  carried out by the University of Bristol BluePebble high performance computing facility. FLEXPART simulations were carried out at the Swiss National Supercomputing Centre (CSCS) under project ID s862. LMW received funding from the European Union's Horizon 2020 research and innovation programme under the Marie Słodowska-Curie grant agreement No. 101030750. LMW and MR were also supported by Natural Environment Research Council (NERC) grants NE/N016548/1 and R100529-101. AJM and ALR are supported by The Met Office Hadley Centre Climate Programme, funded by the UK's Department for Business, Energy and Industrial Strategy and Department for 
475  Environment, Food and Rural Affairs. SP and HP are supported by the National Research Foundation of Korea (NRF) grant funded by the Korean government (MSIT) (no. 2020R1A2C3003774). LJMK is supported by the Ozone Secretariat in Nairobi, which is partly funded for supporting this work by grants from the European Commission, Brussels.



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
