# Peer review of "A renewed rise in global HCFC-141b emissions between 2017-2021"

_Atmospheric Chemistry and Physics, 2022_

## Author Comment (AC1)

We would like to thank both reviewers for their positive and constructive comments, and the editor for their handling of the manuscript. We have addressed any concerns in full, which are detailed below along with other suggested additions and changes.

**Reviewer #1**

Specifics comments:

L11: The word "Therefore" does not following logically from the previous sentences. I suggest to rephrase the sentence or simply omit the word.

We have removed 'therefore' here.

L74: Remove the first occurrence of "not" in the sentence.

This has been removed.

L140-144: Emission release fractions are determined using a statistical approach. Can you explain how you distinguish between A5 and nonA5 countries?

As stated in L138, we use "separate consumption data for Article 5 and non-Article 5 countries, which each have their own prompt and bank release fractions (rather than a single value as used in Simmonds et al. (2017))." We have inserted the additional clause ".., each with very different consumption patterns over time, which allows each to have their own... " for clarity.

It says "as these measurements predate the non-negligible global consumption". Does that mean that release fractions for nonA5 are determined mostly from the period 1990-2000 and for A5 after about 2010?

Here we mean that AGAGE measurements predate the non-negligible global consumption, whereas the NOAA measurements do not. We have clarified this in the text. As the estimate requires an estimate of the bank size, the estimate is much improved by starting from when the bank size is effectively zero.

Also, you assume that the release fractions are constant in time. This seems valid for the 'regular' use of HCFC-141b, but what if there is illegal production, use, or disposal? Please discuss this, maybe around lines L312-313.

This is addressed at L311: "An alternative conclusion ... is that the emissions rate from the bank has increased in more recent years, meaning that the inferred emissions rate, which is constant in time, is no longer representative and cannot adequately model emissions." We have therefore made no changes to the manuscript.

L286-287: The hemispheric differences increases from 2018 to 2021. The low hemispheric difference is only seen for 2018, a single year (what I deduce from Fig 3). With a decrease in use and emissions you would expect the hemispheric difference to become smaller, but

since it is now only seen in one year, I doubt it is a valid/strong argument. Please say something about this.

Following a suggestion from the other reviewer, we have included an extra panel in Figure 3 showing the interhemispheric difference. The N-S difference is now shown in more detail, which makes it clear that the interhemispheric difference follows the growth rates through this period, suggesting that the trend is not anomalous nor an error, and that our conclusions are robust.

P12: Caption Fig 4: From the caption it now seems that the left panel shows the consumption data and not emissions estimated from consumption data. I think you can solve this by writing "..., and from reported consumption data ...".

**Thank you, we have changed this as suggested.**

L330: The disposal could on average be 15 years after peak in consumption, but it will be a rather broad peak, I assume. I would mention this, since it may be a reason why an increase in HCFC-141b emissions started in 2018 (and not in 2026).

We have added the following to Line 333, "and there is likely considerable uncertainty and variation in the lifecycle of HCFC-141b foam containing appliances, and their disposal practices."

L338-340: I would rephrase this sentence. Something like, "A universal leakage rate of 20% in 2020, compared to 0.1% in 2017 would be needed to explain the observed global increase .in emissions ...".

**Thank you, we have made this change as suggested.**

L352-354 and L357-359: The emissions from eastern China, scaled down from the whole of China, from Fang et al. (2018) show an increase (Fig 5), but the emissions from the whole of China from Fang et al. (2019) show a decrease? This seems inconsistent? Is there separate information for the rest of China that shows an increase?

Fang et al. (2018) and Fang et al. (2019) are separate studies, the first of which is a bottomup projection and the second is based on atmospheric measurements within China. The inconsistency between these estimates is currently stated in L357. We have added some additional information at L355 "The top-down derived emissions trend is also supported by the projected emissions of Wang et al. (2015), although emissions projected by Wan et al. (2009) were expected to peak in 2018."

L357: Please give a reference for the "Act on Rational use ...".

**A reference has been added to the text, and the act corrected to the Home Appliance Recycling Act.**

L416-418: I suggest you mention here that the increase in Chinese CFC-11 emissions could only explain 40-60% of the global increase. This would support the idea that the extra

emissions in part originate from regions not monitored. This connect to the statement in L440-442.

We have added at L418 "of which 60±40% could be attributed to eastern China (Park et al. 2021).".

**Reviewer #2**

Different time periods are used for assessing changes in emissions (e.g. 2020 emissions relative to 2008 or 2017) which makes it difficult to compare across regions. Is there a reason for this?

Emissions have been estimated using available data periods and therefore cover different time periods. Different reference years are used to capture the 'big picture' trends as well as the data allows, reflecting the different usage and emissions trends in the different countries and regions. We have therefore not made any changes to the manuscript.

The authors note on line 352 that emissions from eastern China have increased from 3.7 to 7.7 Gg/yr between 2008 and 2020. What fraction of the global increase in emissions since 2017 can be attributed to eastern China? This would provide a useful comparison to the 40-60% of the global increase in CFC-11 emissions found in Rigby et al. 2019 attributed to the same region.

We have added the following to the text at L253: "...and an increase of 1.0 (-2.9, 3.8) Gg yr-1 between 2017-2020. The 2017-2020 increase in eastern China could accounts for around a third of the global increase over the same period  $(3.0 \pm 1.2 \text{ Gg yr}^{-1}, \text{NOAA} \text{ and AGAGE} mean)$ , but given the large uncertainties this could could explain all or none of the global increase." Note that the 2020 emissions estimate in East Asia have been slightly revised, although this makes no different to the conclusions drawn.

Equation 3 – this should be the square root of the sum of squares of uncertainties, no?

Yes, this is correct. Equation 3 in the manuscript was incorrect and has been changed. However, on double checking, a different small mistake was spotted, which has slightly altered the uncertainty estimates for top-down US emissions.

Line 283 – 284 – this increase is being driven by increases in the NH? This does not come through in the figures. I think it would be nice to add a third row to figure 3 showing the hemispheric differences so that this point comes through.

Thank you for this useful suggestion. We have included an extra panel in Figure 3 showing the interhemispheric difference.

Lines 291-293 – why is aggregated emissions here so much higher than the 3 Gg/yr reported in the conclusions over the same period? (Line 415)

L291-293 we report the increase between 2017-2021. In the conclusions we compare the regional emissions to global emissions over the period 2017-2020, as regional estimates for all regions were only available until 2020.

Line 424 – This line does not convey that while emissions from China cannot explain all of the rise in the emissions, it does explain part of the increase.

We have changed this to "...emissions from China alone cannot fully explain the global rise."

Line 394 – This conclusion regarding the increase in US emissions is not clear based on the figure since the observationally-derived emissions are over such a short time period.

It is true that the measurement derived emissions are over a short period of time. The discussion in the paragraph is related to the bottom-up modelled emissions (i.e., the green line in Figure 6a), which is shown for the full time series 1990-2020.

Figure 3 – explain the shaded region around global values here.

Thank you, we have added the following to the figure caption, "The grey shading shows the one standard deviation estimate in the global mean mole fraction".

Figure 4 – is reported consumption data the same thing as bottom-up estimates? Please be consistent throughout for clarity.

We have addressed this following a comment from the other reviewer. The caption now reads "...[derived] from reported consumption data..."

Figure 6 – why does the US bottom-up estimate not have uncertainties but the Australian estimate does?

US bottom-up estimates have been provided by the US EPA (see Sect. 3.4), for which uncertainties are not available. Australian bottom-up emissions have been derived in this study (c.f. L406) using reported consumption and our derived non-Article 5 emissions factor, and its uncertainty.

Minor:

Line 74 – an extra 'not' is included here. Fix typo.

Thanks, fixed.

Line 382 – Add the S to HYSPLIT.

Thanks, fixed.